# Diversity of Volatile Compounds in Ten Varieties of Zingiberaceae

**DOI:** 10.3390/molecules27020565

**Published:** 2022-01-17

**Authors:** Weiyao Peng, Ping Li, Ruimei Ling, Zhenzhen Wang, Xianhui Feng, Ju Liu, Quan Yang, Jian Yan

**Affiliations:** 1Key Laboratory of Agro-Environment in the Tropics, Ministry of Agriculture and Rural Affairs/Guangdong Provincial Key Laboratory of Eco-Circular Agriculture/Guangdong Engineering Research Centre for Modern Eco-Agriculture/College of Natural Resources and Environment, South China Agricultural University, Guangzhou 510642, China; pengweiyao2021@163.com (W.P.); liping2016@scau.edu.cn (P.L.); ruimeiling0725@163.com (R.L.); zhenwang6869@163.com (Z.W.); 2School of Architecture, South China University of Technology, Guangzhou 510641, China; 3Guangdong Provincial Tobacco Shaoguan Co., Ltd., Shaoguan 512000, China; s.asuka@126.com; 4Guangdong Provincial Research Center on Good Agricultural Practice & Comprehensive Agricultural Development Engineering Technology of Cantonese Medicinal Materials/Comprehensive Experimental Station of Guangzhou, Chinese Material Medica, China Agriculture Research System (CARS-21-16)/Key Laboratory of State Administration of Traditional Chinese Medicine for Production & Development of Cantonese Medicinal Materials/School of Traditional Chinese Medicine, Guangdong Pharmaceutical University, Guangzhou 510006, China; yangquan7208@vip.163.com

**Keywords:** Zingiberaceae plants, GC–MS analysis, volatiles, pharmacological effects

## Abstract

Zingiberaceae plants are distributed in the tropical and subtropical regions of the world, being used in many famous medicinal materials. Meanwhile, some Zingiberaceae plants are important horticultural flowers because they are green all year round and have special aromas. To conduct an extensive investigation of the resources of Zingiberaceae plants, the volatile compounds of ten species of Zingiberaceae were extracted and analyzed by GC–MS, including *Costus comosus* var. *bakeri* (K.Schum.) Maas, *Curcuma rubescens* Roxb., *Curcuma aeruginosa* Roxb., *Curcuma attenuata* Wall., Hongfengshou, *Hedychium coronarium* Koeng, *Zingiber zerumbet* (L.) Smith, *Hedychium brevicaule* D. Fang, *Alpinia oxyphylla* Miq., and *Alpinia pumila* Hook.F. A total of 162 compounds were identified, and most of those identified were monoterpenes and sesquiterpenes. (*E*)-labda-8(17),12-diene-15,16-dial, n-hexadecanoic acid, 4-methoxy-6-phenethyl-2H-pyran-2-one, and L-β-pinene were found in high concentrations among the plants. These ten species of Zingiberaceae contained some of the same volatiles, but their contents were different. Pharmacological effects may be associated with the diversity of volatiles in these ten plants.

## 1. Introduction

The family Zingiberaceae comprises about 1500 species that are widely distributed in the tropical and subtropical regions of the world [1]. Zingiberaceae is a family of plants with high medicinal value, which are widely used for the treatment of various diseases: the rhizomes of *Curcuma longa*, *Boesenbergia rotunda*, *Alpinia galanga*, and *Zingiber officinale* are typically used to treat diarrhea, stomachache, and flatulence, for example [2,3,4]. The vast majority of Zingiberaceae plants are both spices and medicinal materials [5].

Many of them are considered important horticultural flowers as well as ornamental plants, or used as raw materials for the production of dyes, food, spices, and fragrances [6]. The stress resistance of *Costus comous* var. *bakeri* (K. Schum) Maas is strong, with fresh cut flowers offering a high yield, good quality, and easy cultivation [7]. The leaves of *Alpinia pumila* Hook.F. have beautiful stripes, and the fruits are bright red when ripe, and they are highly ornamental [8]. *Curcuma attenuate* Wall. is a perennial bulbous herb flower of *Curcuma* in Zingiberaceae, and is a new multi-purpose flower with ornamental and edible value [9]. *Zingiber zerumbet* (L.) Smith is a good garden material and a novel cut flower material, and has ornamental and medicinal value [10]. The rhizome of *Zingiber zerumbet* (L.) Smith can dispel wind and detoxify, treat stomachache and diarrhea, and extract aromatic oil as a blending essence. Its tender stems and leaves can be used as vegetables [11,12]. After screening, excellent lines of ‘Hongfengshou’ in a moderate volume and with abright inflorescence color have been obtained through hybridization, with *Alpinia zerumbet* (Pers.) Burtt. et Smith being used as the female parent and *Alpinia henryi* K. Schumann as the male parent [13]. *Alpinia oxyphylla* Miq. has the effect of warming the spleen, stopping diarrhea and causing salivation [14]. *Curcuma aeruginosa* Roxb. can be used for blood stasis, chest pain, and food retention pain, and may be used against various bacterial and fungal species [15,16]. *Curcuma rubescens* Roxb. can promote blood circulation and relieve pain, depression, and the symptoms of hypoxia, as well as clear the heart and cool the blood [17]. Different extracts of *Hedychium coronarium* Koeng have significant analgesic and anti-inflammatory properties [18]. The rhizome of *Hedychium brevicaule* D. Fang, as a medicinal material, can be used for coughs, phlegm, and asthma [19]. Pictures of all the mentioned plants are shown in Figure 1.

Floral color, longevity, form, and fragrance are significant ornamental characteristics that enhance the esthetic value of ornamental plants [20,21]. These ten plants are of high ornamental value, with graceful leaves, strange flowers, and various colors. They are easy to cultivate, suitable for planting in forests and at watersides, and, due to their distribution in tropical and subtropical areas, suitable for promotion and application in South China. Medicinal Zingiberaceae plants contain a diversity of volatile compounds with various healthcare functions and pharmacological values [22]. However, the volatile components of horticultural Zingiberaceae plants are seldom considered for pharmacology or healthcare. So, ten varieties of Zingiberaceae were investigated, showing a diversity of volatile compounds.

## 2. Results

### 2.1. The Chemical Compositions of Volatiles from Ten Zingiberaceae Plants

As shown on the GC–MS TIC (Figure 2), the volatile chemical constituents of ten *Zingiberaceae* plants were identified using the NIST 2014 database, and the values of the main functional compounds from different plants were described.

The main volatile compound of *C. rubescens* was (*E*)-labda-8(17),12-diene-15,16-dial (11.45%) (Appendix A). Another plant of the same genus is *C. attenuata*, containing 4-methoxy-6-phenethyl-2H-pyran-2-one (100%) and 5,6-dehydrokavain (19.34%) as its main compounds (Appendix A). On the contrary, cyclofenchene (33.83%), L-β-pinene (65.85%), and (*E*)-labda-8(17),12-diene-15,16-dial (94.17%) were detected as the main volatiles of *C. aeruginosa* (Appendix A). Another plant is of the *Costus* genus. The main volatile compounds of *Costus comosus* var. *bakeri* (K.Schum.) Maas were camphene (18.11%), L-bornyl acetate (20.72%), and n-hexadecanoic acid (24.78%) (Appendix A). For the *Z. zerumbet* plant, few volatile compounds were identified, the main compound being myrtenal (21.42%) (Appendix A). Hongfengshou is a new variety obtained by crossing *A. zerumbet*, as the female parent, and *A. henryi*, as the male parent. The main volatile of Hongfengshou was 4-methoxy-6-phenethyl-2H-pyran-2-one (100%) (Appendix A).

Most plants of the genus *Alpinia* are important spices, but their flowers are beautiful as horticultural plants. The main volatiles of *A. oxyphylla* were L-β-pinene (32.06%), trans-sabinene hydrate (32.06%), and cyclofenchene (25.47%). *A. oxyphylla* had an abundance of in (*E*)-labda-8(17),12-diene-15,16-dial (14.05%) (Appendix A). The main volatiles of *A. pumila* were (*E*)-labda-8(17),12-diene-15,16-dial (9.22%), β-pinene (8.32%), n-hexadecanoic acid (6.03%), and (+)-2-bornanon (5.60%). *A. pumila* had many volatile compounds, but their contents were not high (Appendix A).

The main volatile of *H. brevicaule* was L-β-pinene (29.81%), while cyclofenchene (16.21%) was the second-most abundant compound in *H. brevicaule* (Appendix A). The main volatile of *H. coronarium* was germacrone (100%) (Appendix A). 

### 2.2. Diversity of Chemical Characteristics of Ten Zingiberaceae Plants

Some obvious signal peaks were marked in the mass spectra of the different plants. The volatile compound (*E*)-labda-8(17),12-diene-15,16-dial was found in *C. comosus* var. *bakeri*, *C. rubescens*, *C. aeruginosa.*, *H. brevicaule*, *A. oxyphylla*, and *A. pumila.* 4-methoxy-6-phenethyl-2H-pyran-2-one is the main volatile of *C. attenuata* and Hongfengshou (Figure 2). N-hexadecanoic acid was present in all ten species of Zingiberaceae, but there were differences in its content. Its content was higher in *C. comosus*, *C. attenuata*, *A. pumila*, *A. oxyphylla*, and *H. brevicaule*, but less in the other plants (Figure 2).

*C. attenuate* had the highest content of 5,6-dehydrokavain and 4-methoxy-6-phenethyl-2H-pyran-2-one, and 4-methoxy-6-phenethyl-2H-pyran-2-one was also abundant in Hongfengshou. *C. attenuate* and Hongfengshou both contained 5,6-dehydrokavain, 1,3,3-trimethyl-acetate, 1,7,7-trimethylbicyclo[2.2.1]hept-5-en-2-one, 4-methoxy-6-phenethyl-2H-pyran-2-one, and n-hexadecanoic acid. Hongfengshou, *C. aeruginosa*, and *A. oxyphylla* all contained L-β-pinene, cyclofenchene, and (*E*)-labda-8(17),12-diene-15,16-dial (Figure 3). In comparison, the contents of cyclofenchene were higher in *C. aeruginosa*, *A. oxyphylla*, and *H. brevicaule* than others. The contents of (*E*)-labda-8(17),12-diene-15,16-dial were much higher in *C. aeruginosa* and *C. comosus* than others. 

From Figure 2, the species of volatiles in Zingiberaceae were varied. Different species of Zingiberaceae plants contained different volatiles. *A. pumila* showed the most variety of volatile compounds, while *C. attenuate* showed the least variety (Figure 4). A common compound that was detected across all ten plants was n-hexadecanoic acid (Figure 4), which may be the most important and indispensable compound in these plants. Its content was highest in *C. comosus*, so this plant could be considered the most important in terms of relevant composition.

As the most commonly identified compounds were monoterpenes and sesquiterpenes, the proportions terpenoids and others that were detected in the extracts were compared (Figure 5). With the exceptions of *C. attenuate* and Hongfengshou, terpenoids dominated in these plants. They were found at levels of 95% in *C. aeruginosa* and *H. Coronarium*, 92% in *A. oxyphylla*, 84% in *H. brevicaule*, 82% in *C. comosus* and *Z. Zerumbet*, 74% in *C. rubescens*, and 72% in *A. pumila*.

## 3. Discussion

The vast majority of Zingiberaceae plants are not only spices, but also medicinal materials, with a wide range of pharmacological values [23]. The volatile substances of Zingiberaceae plants play an important role. (*E*)-labda-8(17),12-diene-15,16-dial can be used in antibiotics, a-glucosidase inhibition, and anti-fungal treatment [24,25]. It has also been demonstrated that bioactive compounds in (*E*)-labda-8(17),12-diene-15,16-dial can have cytotoxic effects against six human cancer cell lines [26]. L-β-pinene is an important compound for the synthesis of various spices; camphor; vitamins A, E, and K; terpene resin; etc. These can be used as chemical reagents, fine chemicals, pharmaceutical intermediates, and material intermediates [27]. Camphene can be used to synthesize perfume, pesticides, camphor, insecticides, borneol, etc. [28,29]. N-hexadecanoic acid is a kind of white and pearlescent phosphorus, which can be used as a spice ingredient because of its special aroma and taste [30]. In addition, it has been reported to have had anti-inflammatory effects and can be used for balancing glycemic levels [31,32]. Germacrone is a monocyclic sesquiterpene existing in Zingiberaceae plants. A lot of research has shown that germacrone can inhibit the proliferation and promote apoptosis of hepatoma cells, breast cancer cells, lung cancer cells, and glioma cells, and has anti-tumor effects in a quality-dependent manner [33,34]. 4-methoxy-6-phenethyl-2H-pyran-2-one is used for chemical reagents, fine chemicals, pharmaceutical intermediates, and material intermediates [35,36]. 5,6-Dehydrokavain is a pyranone compound with biological activity, which can antagonize experimental gastric and duodenal ulcers [37]. Cyclofenchene remarkably inhibits the growth of tested Gram-positive and Gram-negative bacteria, except for *Pseudomonas aeruginosa* [38,39]. 9,12-octadecadienoic acid (Z,Z) may be used for the pharmacological treatment of fever and inflammation [40]. Trans-sabinene hydrate and β-elemene have anti-bacterial and anti-inflammatory activities [41,42]. Sabinene can also be useful for its favorable aroma attributes (citrus, orange peel, acidic) [43]. However, some volatile compounds are present in high concentration in these plants, such as ambrial, 2-methyl-3-phenyl-propanal, and 1,7,7-trimethylbicyclo[2.2.1]hept-5-en-2-one, but pharmacological reports are lacking. The most commonly identified compounds were terpenoids, which have been reported as having pharmacological value [44]. Therefore, pharmacological tests should be carried out in the future. The pharmacological effects of these compounds may be related to those of the plants, but further research is needed to confirm this.

There are several techniques for extracting volatile compounds from plants. Baojun Tu et al. extracted volatile compounds with dichloromethane as the extraction solvent, and analyzed them by GAS chromatography–mass spectrometry (GC–MS) with simultaneous distillation extraction (SDE) [45]. The analysis of X. X. Zhang et al. was performed by headspace solid-phase microextraction gas chromatography–mass spectrometry (HS-SPME–GC–MS) [46]. The analysis of Freitas, T. P. et al. was performed using a headspace solid-phase microextraction combined with gas chromatography–mass spectrometry (SPME–GC–MS) [47]. Different extraction and analysis techniques may result in different volatile compound analysis results, so different extraction and analysis methods can be tried in the later stages.

The rhizome of *C. aeruginosa* has been used as a traditional medicine for gastrointestinal issues such as diarrhea and colic, as well as for uterine involution, uterine pain, and inflammation [48]. *C. attenuata* has shown medicinal value, and most *Curcuma* species have been used as good natural anti-oxidants in functional foods, medicines, and cosmetics [49,50]. Modern pharmacological studies have shown that the fruit of *A. oxyphylla* has many biological activities, such as regulating urination, improving cognitive ability, improving diabetes, and as an anti-bacterial and anti-tumor treatments [51]. *Z. zerumbet* can be used to treat inflammation, fever, toothache, diarrhea, pain, muscles spasms, and rheumatisms [52]. *A. pumila* has been widely used in the treatment of traumatic injury as a folk medicine [53]. According to reports, all the ten Zingiberaceae plants have both medicinal and ornamental value [7,8,9,10,11,12,13,14,15,16,17,18,19]. They are rich in terpenoids, and the structures of their main volatile chemicals are shown in Figure 6. Most of the Zingiberaceae plants are medicinal plants which have unique therapeutic capacities and the ability to treat various human diseases [54,55,56]. The differences in the contents and compositions of their volatile components leads to differences in aroma and medicinal value among different plants [57].

There are many methods to extract volatile oil in the early stages, such as water distillation, solvent extraction, headspace microextraction, and so on. Because different extraction methods have certain influences on the main components of volatile oils, dichloromethane was selected as the solvent for extraction, considering the short time and simple operation of the solvent extraction method. This study compared the differences of volatile components in ten Zingiberaceae plants. According to the analysis results, the volatile compounds of these plants of different genera differ greatly, and the volatile compounds from the same genus also differ greatly. The purpose of this study has been to provide a reference for other research on the development and utilization of plants.

## 4. Materials and Methods

### 4.1. Plant Materials

Ten horticultural flower species varieties (Figure 1) were provided by the PB company in 2018, and were identified by associate professor Xianhui Feng at SCUT. Plant materials were dried at room temperature (25 °C) to a consistent weight and stored in a refrigerator at −20 °C for later use.

### 4.2. Extraction of Volatile Oil

About 100 mg of each dried plant sample was immersed in a moderate amount of liquid nitrogen, powdered by a plant mill (Xuman 2500 g, Jinhua, China), and placed in a 1.5 mL HPLC sample bottle. Chromatographic pure hexane (containing 20 mg/L cyclohexanone, internal standard) was added into the sample bottle and was extracted with a concentration of 100 mg/mL, repeated three times for each dry sample. After ultrasonic (10L touch degassing + 240 W ultrasonic + 450 W heating, Shenzhen, China) treatment at room temperature (25 °C) for 30 min, it was allowed to stand for 1 h, then added to an appropriate amount of anhydrous sodium sulfate, sealed, and put in a refrigerator at 4 °C overnight. Next, 200 μL of supernatant was aspirated and transferred to the sample bottle, containing a 200 μL endotracheal tube for GC–MS (Agilent 7890A 5975, Santa Clara, CA, USA; column: Agilent hp-5 19091j-413, Santa Clara, CA, USA) analysis [58].

### 4.3. Gas Chromatography–Mass Spectrometry Conditions

GC conditions: A db-5ms elastic quartz capillary column (30 m × 0.25 mm × 0.25 μm) was used; the initial temperature of the column was 60 °C, which was maintained for 2 min and then increased to 250 °C at a rate of 6 °C/min, and maintained for 15 min. The injection port temperature was 230 °C. The carrier gas was high-purity helium at aflow rate of 1.0 mL/min with no split flow, and the injection volume was 1 μL. MS conditions: An EI ion source was used with an ionization energy of 70 EV, a scanning range of *m*/*z* 29~500 amu, an ion source temperature of 230 °C, and a GC/MS interface temperature of 280 °C. Determination of retention index value: The retention index was determined according to the method of Liang Sheng et al. [59]. Mixed-standard samples of n-alkanes were analyzed by GC–MS under the same conditions (injection volume—1 μL). The retention time of each standard was recorded. The retention index of each component was calculated by the Kovats Retention Index (RI) formula.

Data processing and analysis: The standard mass spectrometry database NIST 2014 was used in the matching comparison to search for data with a matching rate of more than 85%. The relative content of each component was converted by peak area normalization, with cyclohexanone as the internal standard.

## 5. Conclusions

The chemical constituents of ten species of Zingiberaceae were analyzed by GC–MS, finding a rich diversity of chemical structures, and showing that their contents were different from monoterpenes and sesquiterpenes possessing special pharmacological functions. Some plants may be used not only for landscaping and ornamental purposes, but also for healthcare.

## Figures and Tables

**Figure 1 molecules-27-00565-f001:**
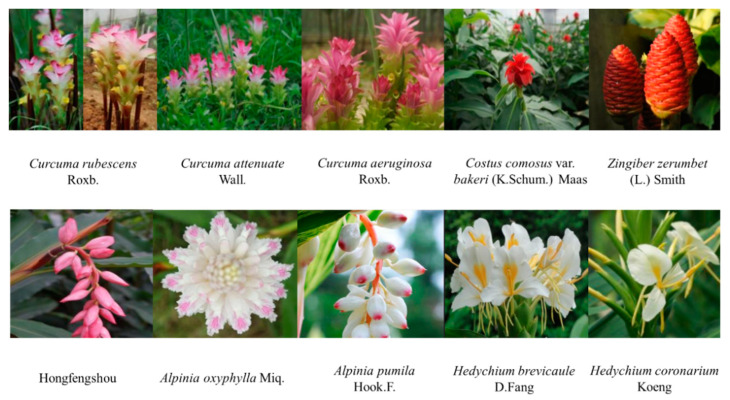
These are pictures of ten species of Zingiberaceae. *Curcuma rubescens* Roxb.; *Curcuma attenuate* Wall.; *Curcuma aeruginosa* Roxb.; *Costus comosus* var. *bakeri* (K.Schum.) Maas; *Zingiber zerumbet* (L.) Smith; a new *Alpinia* cultivar, Hongfengshou; *Alpinia oxyphylla* Miq.; *Alpinia pumila* Hook.F.; *Hedychium brevicaule* D. Fang; and *Hedychium coronarium* Koeng.

**Figure 2 molecules-27-00565-f002:**
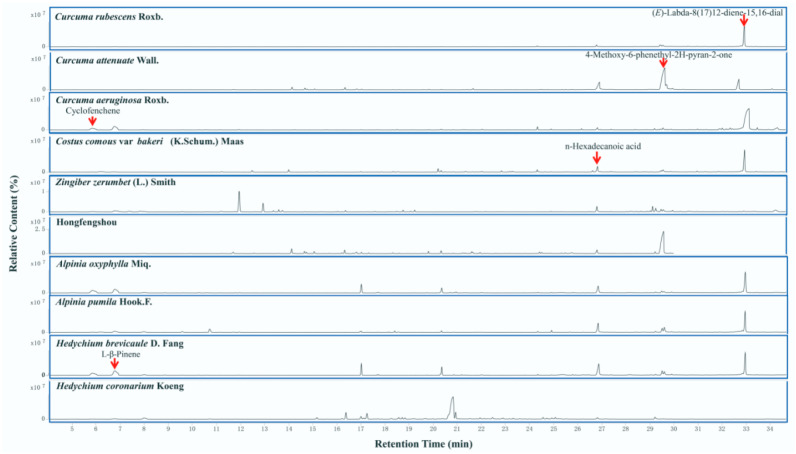
GC–MS TIC of different Zingiberaceae plants. The peaks indicated by the red arrows represent volatiles labeled in the figure. Peaks at the same time indicate the same substance.

**Figure 3 molecules-27-00565-f003:**
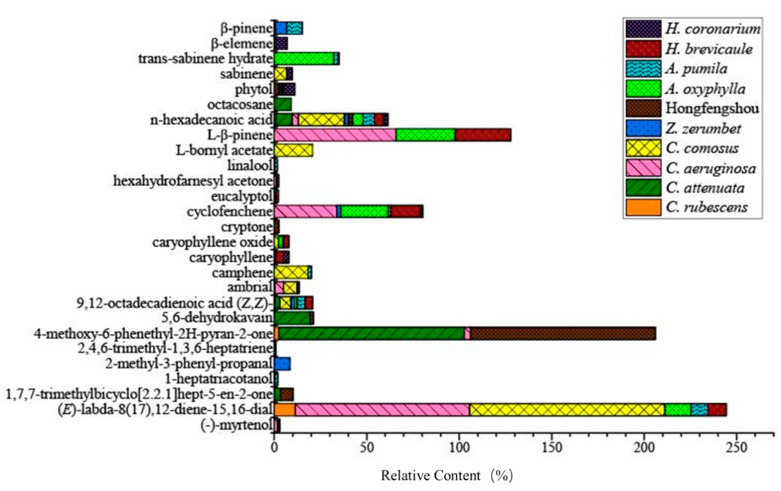
Relative contents of common volatiles in different Zingiberaceae plants. Different color blocks represent different plants. The size of the color block area represents the relative content.

**Figure 4 molecules-27-00565-f004:**
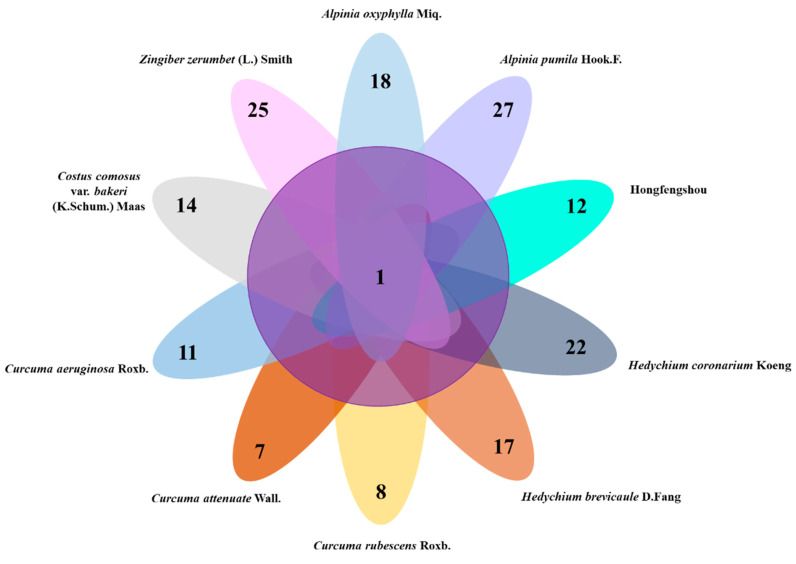
Venn diagram of ten species of Zingiberaceae. The number in the middle represents the number of common volatiles across various Zingiberaceae plants, and the numbers in each circle represents the number of unique volatiles in each Zingiberaceae plant.

**Figure 5 molecules-27-00565-f005:**
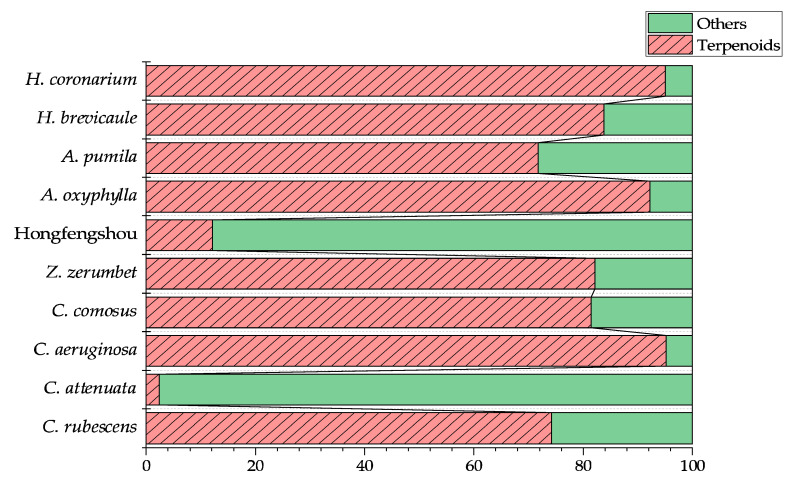
Total amount of terpenoids and others in ten Zingiberaceae plants. The values represent the proportion of terpenoids and others in each of the plants.

**Figure 6 molecules-27-00565-f006:**
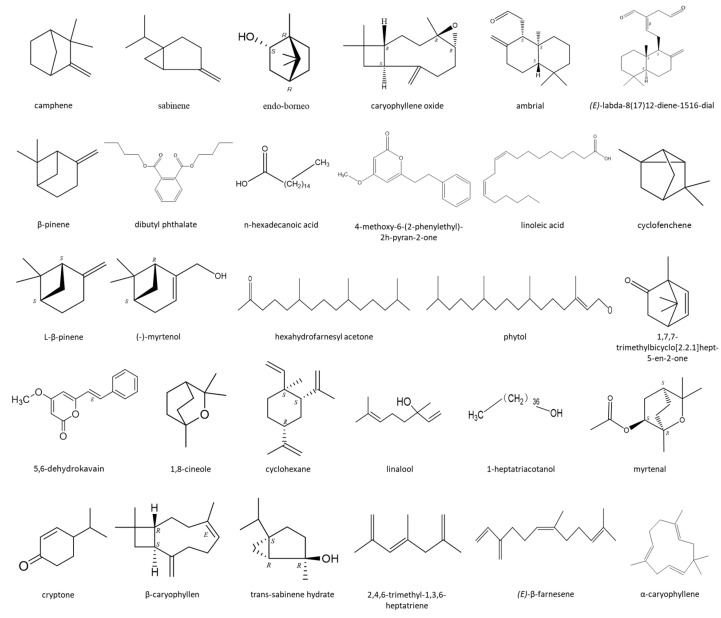
Structural formulae of the main volatiles from ten species of Zingiberaceae plants.

## Data Availability

Data are available in Electronic Supporting Information (ESI), and for additional details, please contact the authors.

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
