# Peer review of "Diversity of Volatile Compounds in Ten Varieties of Zingiberaceae"

_molecules, 2022, doi:10.3390/molecules27020565_

Round 1

Reviewer 1 Report

General comments

The paper brings important information about the composition of different Zingiberaceae plants through the identification of their volatile compounds. However, the novelty of the paper is not so highlighted in the text and should be improved. The text brings comparison among different plants but a better information explaining why those ones were chosen would improve the relevance of the paper. As the most identified compounds were monoterpenes and sesquiterpenes, maybe a comparison among the total amount of these groups in the extracts could give a quickly visual information regarding differences on plants composition. So, the questions were pointed in the text. Further, I recommend pointing the most relevant plant, in terms of volatile compounds composition, among the plants evaluated.

Comments:

1) Page 2, line 76: “However, the volatile components of horticultural plants of Zingiberaceae are seldom noticed as the pharmacological value of health care”. As the authors mentioned the pharmacological value of volatile compounds, I was expecting a biological activity evaluation of the mainly compounds extracted from those plants. So, I suggest that the authors list the main compounds extracted from Zingiberaceae which they recommend pharmacological test that can be carried out as future perspectives, or further works to be evaluated.

2) Page 2, line 77: The objective of the work was not so clear. The authors mentioned the varieties of Zingiberaceae and their characteristics, but once they reinforce the high diversity of these kind of plants, I suggest the authors to improve the sentence explaining why those specific varieties were chosen. Further, I suggest a short paragraph explaining the state of the art of the technique used to characterize the volatile compounds, once in the conclusion they mentioned “The purpose of this study is to provide a reference for other research methods and development and utilization of undergraduate plants.” (Page 12, line 187)

3) Page 3, line 104: “The volatiles contents of Z. zerumbet are not too much, the myrtenal (21.42%) was the main volatile (Table 1) ”.. Consider changing this sentence to “For Z.  zerumbet plant it was not identified many volatile compounds, being the myrtenal (21.42%) the main volatile compound identified (Table 1).”

4) Page 4, line 114: By “rich” the authors mean “diversity” or “compounds of high value/significance”? It was not clear; I suggest for them to improve the sentence.  

5) Page 4, Table 1: The degree of matching/probability in percentage for each compound were not available, even pointed in the table (“Compound name (Match/%)”). I suggest to authors to insert the value. Further, the description of the table should be improved to become more information regarding its content.

6) Page 9, line 127: “different the plants”.. change to “different plants”

7) Page 9, line 131: “N-hexadecanoic acid existed..” change to “N-hexadecanoic acid was present..and change “…were differences in content”  to  “..were differences in the content”.

8) Page 9, line 132: “but less in other plants” change to “but less in the other plants.”

9) Page 9, line 135: “were also abundant” change to “was also abundant.”

10) Page 10, line 152: “From Fig. 2, the species and quantity of volatiles in Zingiberaceae were very differently. Different species of Zingiberaceae plants contained different volatiles. A. pumila contained the most volatile compounds, while C. attenuate contained the least (Fig. 4).”

Observing the figure 2, I think that the comparison is between the variety of compounds and no between the content. A. pumila showed the most variety of volatile compounds, while the C. attenuate showed the lowest variety. Is that what the authors wanted to mean with the sentence? If positive, I suggest to improve the sentence for a better explanation.

11) Page 11, figure 4: Which was the most common compound found among Zingiberaceae plants? And which one do you consider as the most important compound? Which plant could be considered as the most important in terms of relevant composition?

12) Page 11, line 168: “In this study, all the ten Zingiberaceae plants had both medicinal and ornamental value.” This affirmation is based on their results or get from another research? It is not so clear. I suggest to the authors to provide a reference at this point to improve and support the sentence.

13) Page 12, line 187: “The purpose of this study is to provide a reference for other research methods and development and utilization of undergraduate plants”. Once the authors mentioned that the purpose of the study was to provide a reference of methods for other research they might discuss or compare the methods used with other methods in literature.

14) Discussion: What about the commercial value of Zingiberaceae plants? The authors mentioned the pharmacological potential of those plants, but I would like to ask about their opinion regarding the amount of those compounds in the plants, and the if they think that if is worth to product those plant in large scale to extracted them. Can one of the plants be considered as a potential source?

15) Page 12, line 184: “According to the analysis results, the volatile compounds of zingiberaceae plants of different genera differ greatly, and the volatile compounds of zingiberaceae plants of the same genus also differ greatly”.  Did the authors made a statistical analysis among the relative contents of common volatiles in different Zingiberaceae plants to identify statistically significant differences?

16) Page 13, line 223: The use for pharmacological purpose is a possibility, right? The author should mention which plants (as examples among the pants evaluated) can be used for pharmacological purpose.

17) Page 13, line 223: “Some plants maybe be used as health care except of greening and viewing.” The meaning of the sentence is not so clear, please consider change the word “except”.

Author Response

Response to Reviewer 1 Comments

The paper brings important information about the composition of different Zingiberaceae plants through the identification of their volatile compounds. However, the novelty of the paper is not so highlighted in the text and should be improved. The text brings comparison among different plants but a better information explaining why those ones were chosen would improve the relevance of the paper. As the most identified compounds were monoterpenes and sesquiterpenes, maybe a comparison among the total amount of these groups in the extracts could give a quickly visual information regarding differences on plants composition. So, the questions were pointed in the text. Further, I recommend pointing the most relevant plant, in terms of volatile compounds composition, among the plants evaluated.

 >> Page3 Line77-80: Added the reasons why these plants were selected. The total amounts of terpenoids and others were compared in the newly added Figure 5. These plants were uniformly evaluated in volatile composition.

Comments:

1) Page 2, line 76: “However, the volatile components of horticultural plants of Zingiberaceae are seldom noticed as the pharmacological value of health care”. As the authors mentioned the pharmacological value of volatile compounds, I was expecting a biological activity evaluation of the mainly compounds extracted from those plants. So, I suggest that the authors list the main compounds extracted from Zingiberaceae which they recommend pharmacological test that can be carried out as future perspectives, or further works to be evaluated.

 >> Main compounds and prospects for future pharmacological testing were listed in the discussion.

2) Page 2, line 77: The objective of the work was not so clear. The authors mentioned the varieties of Zingiberaceae and their characteristics, but once they reinforce the high diversity of these kind of plants, I suggest the authors to improve the sentence explaining why those specific varieties were chosen. Further, I suggest a short paragraph explaining the state of the art of the technique used to characterize the volatile compounds, once in the conclusion they mentioned “The purpose of this study is to provide a reference for other research methods and development and utilization of undergraduate plants.” (Page 12, line 187)

 >> Page3 Line77-80: Added the reasons why these plants were selected. Page16 Line235-244: The state of the art of the technique used to characterize the volatile compounds was explained.

3) Page 3, line 104: “The volatiles contents of Z. zerumbet are not too much, the myrtenal (21.42%) was the main volatile (Table 1) ”.. Consider changing this sentence to “For Z.  zerumbet plant it was not identified many volatile compounds, being the myrtenal (21.42%) the main volatile compound identified (Table 1).”

 >> It has been changed.

4) Page 4, line 114: By “rich” the authors mean “diversity” or “compounds of high value/significance”? It was not clear; I suggest for them to improve the sentence.  

 >> It meant “compounds of high value” and has been changed.

5) Page 4, Table 1: The degree of matching/probability in percentage for each compound were not available, even pointed in the table (“Compound name (Match/%)”). I suggest to authors to insert the value. Further, the description of the table should be improved to become more information regarding its content.

>> The degree of matching/probability in percentage for each compound is the relative amount compared to cyclohexanone. Relative content = peak area of substance/peak area of cyclohexanone ×10

6) Page 9, line 127: “different the plants”.. change to “different plants”

>> It has been changed.

7) Page 9, line 131: “N-hexadecanoic acid existed..” change to “N-hexadecanoic acid was present..” and change “…were differences in content”  to  “..were differences in the content”.

 >> It has been changed.

8) Page 9, line 132: “but less in other plants” change to “but less in the other plants.”

 >> It has been changed.

9) Page 9, line 135: “were also abundant” change to “was also abundant.”

 >> It has been changed.

10) Page 10, line 152: “From Fig. 2, the species and quantity of volatiles in Zingiberaceae were very differently. Different species of Zingiberaceae plants contained different volatiles. A. pumila contained the most volatile compounds, while C. attenuate contained the least (Fig. 4).”

Observing the figure 2, I think that the comparison is between the variety of compounds and no between the content. A. pumila showed the most variety of volatile compounds, while the C. attenuate showed the lowest variety. Is that what the authors wanted to mean with the sentence? If positive, I suggest to improve the sentence for a better explanation.

>> It has been changed.

11) Page 11, figure 4: Which was the most common compound found among Zingiberaceae plants? And which one do you consider as the most important compound? Which plant could be considered as the most important in terms of relevant composition?

>> N-hexadecanoic acid was the most common compound found among these plants. N-hexanoic acid is probably the indispensable compound in zingiberaceae. Costus comosus var. bakeri (K.Schum.) Maas contained the the highest content of n-hexadecanoic acid, which is the most important plant.

12) Page 11, line 168: “In this study, all the ten Zingiberaceae plants had both medicinal and ornamental value.” This affirmation is based on their results or get from another research? It is not so clear. I suggest to the authors to provide a reference at this point to improve and support the sentence.

>> Page 12, line 254: It has been added.

13) Page 12, line 187: “The purpose of this study is to provide a reference for other research methods and development and utilization of undergraduate plants”. Once the authors mentioned that the purpose of the study was to provide a reference of methods for other research they might discuss or compare the methods used with other methods in literature.

>> It would be a pleasure that they discuss or compare the methods used with other methods in literature.

14) Discussion: What about the commercial value of Zingiberaceae plants? The authors mentioned the pharmacological potential of those plants, but I would like to ask about their opinion regarding the amount of those compounds in the plants, and the if they think that if is worth to product those plant in large scale to extracted them. Can one of the plants be considered as a potential source?

>> It is worth to product those plant in large scale not only for the pharmacological potential but also for the ornamental and economic value. Many terpenoids with pharmacological effects and several compounds such as 4-methoxy-6-phenethyl-2h-Pyran-2-one and N-hexadecanoic acid are abundant in these plants. Further research is needed on which plants can be considered as potential sources.

15) Page 12, line 184: “According to the analysis results, the volatile compounds of zingiberaceae plants of different genera differ greatly, and the volatile compounds of zingiberaceae plants of the same genus also differ greatly”.  Did the authors made a statistical analysis among the relative contents of common volatiles in different Zingiberaceae plants to identify statistically significant differences?

>> The relative contents of common volatiles in different zingiberaceae plants were not analyzed statistically.

16) Page 13, line 223: The use for pharmacological purpose is a possibility, right? The author should mention which plants (as examples among the pants evaluated) can be used for pharmacological purpose.

>> Pharmacological purposes are certainly a possibility. All of these plants can be used for pharmacological purposes.

17) Page 13, line 223: “Some plants maybe be used as health care except of greening and viewing.” The meaning of the sentence is not so clear, please consider change the word “except”.

>> It has been changed.

Reviewer 2 Report

I think that the manuscript entitled “Diversity of volatile compounds in ten varieties of Zingiberaceae" deserves publication in after major revision. Interesting work, lack of discussion and statistic.

Line 31-32: it has not been tested

Line 37-39 or 45-46: please delete sentence “Many of them are considered important horticultural flower as ornamental plants or used as raw materials for the production of dyes, food, spices and fragrances [3].”

Line 63: please change “hypopepsia” into „hypoxia”

Line 68: These are own photos?

Line 95-96, 99-103, 115-117, 119-122, 138-140, 144-147, 159-173: This is not results maybe introduction or discussion?

Line 123: please change “min” into “min.”

Line 123: whether instead of “0” in the Table 1 there should not be “nd” (not detection)

Table 1 it is hardly legible

Line 133: please change “plants. (Fig. 2).” Into “plants (Fig. 2).”

Line 145: please change “Pseudomonas aeruginosa” into “Pseudomonas aeruginosa

Line 165: delete “antibacterial, anti-tumor”

Line 177, 178, 184, 185, 186: Please change “zingiberaceae” into ”Zingiberaceae”

Line 177-188: a lot of repetition “zingiberaceae plants”

Line 177-188: This is not discussion

Line 190: How the plant materials was prepared/preserved include the use of devices (dried, powdered)

Line 195: please change “mg / L” into “mg/L”

Line 197, 206: please change “ml” into “mL”

Line 197: please change “mg / ml” into “mg/ml”

Line 197: please complete on the information about the device ultrasonic incl. name, type, city, country

Line 198: please complete at room temperature information

Line 200, 201: please change “μl” into “μL”

Line 202: please complete on the information about gas chromatography-mass spectrometry, column etc. incl. name, type, city, country

Line 205: please change ”6℃ / min” into “6℃/min”

Line 208: please change “GC / MS” into “GC/MS”

Line 210: please change “al” into “al.”

Line 219: it was known before reading the manuscript

Line 234: only two references from 2020 and two references from 2019 the rest are older

Author Response

I think that the manuscript entitled “Diversity of volatile compounds in ten varieties of Zingiberaceae" deserves publication in after major revision. Interesting work, lack of discussion and statistic.

>> We have revised the manuscript. And the discussion has been changed.

Line 31-32: it has not been tested

>> It has been changed.

Line 37-39 or 45-46: please delete sentence “Many of them are considered important horticultural flower as ornamental plants or used as raw materials for the production of dyes, food, spices and fragrances [3].”

>> It has been changed.

Line 63: please change “hypopepsia” into „hypoxia”

>> It has been changed.

Line 68: These are own photos?

>> Figure 1 has been replaced with our own photos.

Line 95-96, 99-103, 115-117, 119-122, 138-140, 144-147, 159-173: This is not results maybe introduction or discussion?

>> It has been changed in discussion.

Line 123: please change “min” into “min.”

>> It has been changed.

Line 123: whether instead of “0” in the Table 1 there should not be “nd” (not detection)

>> It has been changed.

Table 1 it is hardly legible

>> Table 1 has been changed in the supplementary materials.

Line 133: please change “plants. (Fig. 2).” Into “plants (Fig. 2).”

>> It has been changed.

Line 145: please change “Pseudomonas aeruginosa” into “Pseudomonas aeruginosa

>> It has been changed.

Line 165: delete “antibacterial, anti-tumor”

>> It has been changed.

Line 177, 178, 184, 185, 186: Please change “zingiberaceae” into ”Zingiberaceae”

>> It has been changed.

Line 177-188: a lot of repetition “zingiberaceae plants”

>> It has been changed.

Line 177-188: This is not discussion

>> The discussion has been changed.

Line 190: How the plant materials was prepared/preserved include the use of devices (dried, powdered)

>> It has been added.

Line 195: please change “mg / L” into “mg/L”

>> It has been changed.

Line 197, 206: please change “ml” into “mL”

>> It has been changed.

Line 197: please change “mg / ml” into “mg/ml”

>> It has been changed.

Line 197: please complete on the information about the device ultrasonic incl. name, type, city, country

>> It has been added.

Line 198: please complete at room temperature information

>> It has been added.

Line 200, 201: please change “μl” into “μL”

>> It has been changed.

Line 202: please complete on the information about gas chromatography-mass spectrometry, column etc. incl. name, type, city, country

>> It has been added.

Line 205: please change ”6℃ / min” into “6℃/min”

>> It has been changed.

Line 208: please change “GC / MS” into “GC/MS”

>> It has been changed.

Line 210: please change “al” into “al.”

>> It has been changed.

Line 219: it was known before reading the manuscript

>> The sentence  were deleted.

Line 234: only two references from 2020 and two references from 2019 the rest are older

>> It has been changed.

Round 2

Reviewer 2 Report

I think that the re-submitted the manuscript entitled “Diversity of volatile compounds in ten varieties of Zingiberaceae" deserves publication in Molecules after minor revision.

Line 1: please chose “Type of the Paper”

Line 123: whether instead of “0” in the Table 1 there should not be “nd” (not detection)

>> It has been changed.

>>>It has not been changed

Line 275: References not in accordance with the requirements of the Molecules. It applies to all items.

Why has been no statistics? mean±SE or mean±SD?  post-hoc test?

Author Response

Line 1: please chose “Type of the Paper”

>> Article

Line 123: whether instead of “0” in the Table 1 there should not be “nd” (not detection)

>> It has been changed.

>>>It has not been changed

>>>> Now it has been changed in supplementary table1.

Line 275: References not in accordance with the requirements of the Molecules. It applies to all items.

>> The references have been changed to MDPI citation format.

Why has been no statistics? mean±SE or mean±SD?  post-hoc test?

>> The GC-MS data was tested only once.